# Barriers to accessing maternal health care services in the Chittagong Hill Tracts, Bangladesh: A qualitative descriptive study of Indigenous women's experiences

Shahinoor Akter [1,2,3]*, Kate Davies[4‡], Jane Louise Rich[1,5‡], Kerry Jill Inder[6,7‡]

**1** School of Medicine and Public Health, Faculty of Health and Medicine, University of Newcastle, Callaghan, New South Wales, Australia, **2** Hunter Medical Research Institute, New Lambton Heights, New South Wales, Australia, **3** Department of Anthropology, Jagannath University, Dhaka, Bangladesh, **4** School of Humanities and Social Science, Faculty of Education and Arts, University of Newcastle, Callaghan, New South Wales, Australia, **5** Priority Research Centre for Brain and Mental Health, University of Newcastle, Callaghan, New South Wales, Australia, **6** Priority Research Centre for Generational Health and Ageing, University of Newcastle, Callaghan, New South Wales, Australia, **7** School of Nursing and Midwifery, Faculty of Health and Medicine, University of Newcastle, Callaghan, New South Wales, Australia

‡ These authors are co-authors in this work.
* Shahinoor.Akter@uon.edu.au

**Data Availability Statement:** Following ethical approval provided by the Human Research Ethics Committee of the University of Newcastle, Australia, non-identifiable data may be shared with

## Abstract

### Background

Increased maternal health care (MHC) service utilisation in Bangladesh over the past decades has contributed to improvements in maternal health outcomes nationally, yet there is little understanding of Indigenous women's experiences of accessing MHC services in Bangladesh.

### Methods

Face-to-face semi-structured qualitative interviews with 21 Indigenous women (aged 15–49 years) within 36 months of delivery from three ethnic groups (Chakma, Marma and Tripura) were conducted between September 2017 and February 2018 in Khagrachhari district. Purposive sampling was used to recruit women representative of the population distribution in terms of age, ethnic community and service use experience. All interviews were conducted in Bangla language and audio-recorded with consent. Interviews were transcribed directly into English before being coded. Data were analysed thematically using a qualitative descriptive approach aided by NVivo12 software.

### Results

Of the 21 women interviewed, 14 had accessed at least one MHC service during their last pregnancy or childbirth and were categorised as the User group. The remaining seven participants were categorised as 'Non-users' as they had not access antenatal care, facility delivery or postnatal care services. Women reported that they wanted culturally relevant, respectful, home-based and affordable care, and generally perceived formal MHC services

other parties to encourage scientific scrutiny and to contribute to further research and public knowledge. As the study was conducted in a relatively small population area, with a very limited number of health care providers and facilities, full interview transcripts may make some participants, health services or specific locations potentially identifiable. Data can be shared upon request via by contacting NOVA, the University of Newcastle open access repository via Nova@newcastle.edu.au.

**Funding:** This research was supported by an Australian Government Research Training Program (RTP) Scholarship.

**Competing interests:** The authors have declared that no competing interests exist.

as being only for complications and emergencies. Barriers to accessing MHC services included low levels of understanding about the importance of MHC services, concerns about service costs, limited transport and fears of intrusive practices. Experiences within health services that deterred women from accessing future MHC services included demands for unofficial payments and abusive treatment by public facility staff.

## Conclusion

Improving access to MHC services for the CHT Indigenous women requires improved understandings of cultural values, priorities and concerns. Multifaceted reform is needed at individual, community and health systems levels to offer culturally appropriate health education and flexible service delivery options.

## Introduction

Improving maternal health is a global public health priority [1]. Bangladesh, a low and middle-income country, reduced its maternal mortality rate from 322 to 194 deaths per 100,000 live-births between 2001 and 2010 [2]. Findings from large household surveys conducted in 2001, 2010 and 2014 revealed that increased use of maternal health care (MHC) services was a key contributor to this decline [3–5]. The proportion of women in Bangladesh receiving at least one antenatal care (ANC) visit from a medically trained health care provider is 64% and accessing a public or private health facility for childbirth is 37% [6]. However, there is a lack of information regarding MHC accessibility and utilisation for cultural minority groups in Bangladesh, including Indigenous women in the Chittagong Hill Tracts (CHT) [7, 8].

Cultural minority groups form based on characteristics such as religion, race, ethnicity and indigeneity and generally share a cultural identity that is different to the majority population in which they live [9]. Globally, cultural minority groups tend to occupy subordinate positions in multi-cultural societies, experience separation and exclusion and suffer discrimination [10, 11]. Indigenous people identify themselves as being part of a distinct cultural group and are often marginalised populations that live within or attached to an ancestral geographical territory. The cultural and social institutions of Indigenous groups are distinctive from the majority culture with who they share the territory [12]. Bangladesh has 54 Indigenous groups with distinctive social and cultural identities, and eleven Indigenous groups live in Chittagong Hill Tracts (CHT) region [13]. The identity of the Indigenous groups of the CHT has been contested since the colonial period. They were referred to as tribal *or Upojati* (a meaning similar to 'uncivilized') signifying their perceived lower status. Although Indigenous groups of Bangladesh seek constitutional recognition as Indigenous (in Bangla *Adivasi*), they are recognised as "tribal groups" in the health policy of Bangladesh [14–18].

The CHT, in south-eastern Bangladesh, consists of three hill districts: Bandarban, Khagrachhari and Rangamati, home to eleven Indigenous communities [19]. The CHT communities, isolated politically and socially until a Peace Accord was signed in 1997 between the Government of Bangladesh and the Indigenous communities [20], have received minimal policy recognition regarding their human rights, including health rights. While national MHC access rates have improved, the health system has not necessarily provided inclusive or equitable care for women from diverse socio-cultural backgrounds including from Indigenous communities [3, 21]. Like many countries, Bangladesh policy has prioritised the health needs of the

majority Bengali population, and health rights of other cultural minority groups, including Indigenous groups, have not been well-addressed [8, 22, 23].

Limited available literature on the health of CHT Indigenous women suggests accessing MHC services during pregnancy and childbirth is lower among this population, as is birth weight [24–26]. Indigenous women lack knowledge regarding MHC services [27]. National estimates for accessing antenatal, facility delivery and postnatal care services are 78%, 37% and 36% respectively [28] compared to 53%, 33% and 10% respectively for Indigenous women in the CHT [29]. Only 6.4% of Indigenous women accessed all three MHC services (antenatal, facility delivery and postnatal care) for their last pregnancy [29], significantly lower than the national average of 43% [4].

Lack of adequate, accurate data on health and healthcare access is a significant gap for Indigenous communities' worldwide, including Bangladesh [30]. The Sustainable Development Goals, in particular goal 3.1, aim to minimise such gaps for Indigenous populations [31, 32]. Although there is a lack of quality research on Indigenous women's health, the Seventh Five-Year Plan (2016–2020) from the government of Bangladesh indicated the importance of ensuring Indigenous friendly health services [7, 18].

Indigenous people worldwide experience systematic discrimination and exploitation, which becomes two-fold for Indigenous women due to their ethnic and gender identities [23]. Indigenous women in low- and middle-income countries experience significant inequities in accessing MHC services, including Indigenous-focused health intervention programs [33]. Since the 1980s, western healthcare models have acknowledged the importance of community participation and intercultural healthcare approaches to reduce such inequities and inequalities [34, 35]. Countries such as Guatemala, Mexico and Vietnam, have adopted intercultural service provisions for pregnant Indigenous women; however, they earned criticism for not ensuring community engagement [36–38].

Little is known about Indigenous women's experiences of accessing MHC services, suggesting that Indigenous women's health issues have been a low policy and program priority, contributing to this group having the worst health outcomes in Bangladesh [15, 24, 39]. This research aimed to address the lack of data about the health of Indigenous women in the CHT to inform strategies to improve maternal health outcomes and contribute to the Government of Bangladesh's goal to develop Indigenous friendly health services [7]. This qualitative study explored Indigenous women's perspectives on what prevents or deters them from accessing MHC services, and what they value as the important characteristics of MHC. In understanding the barriers to MHC service access, it aimed to better understand culturally appropriate ways to improve these women's access to healthcare.

## Methods

The overview of qualitative methods below has been checked against the Consolidated Criteria for Reporting Qualitative Studies (COREQ) guidelines [40].

### Study design

This was a qualitative study involving in-depth interviews with Indigenous women within 36 months of delivery in Khagrachhari district, Bangladesh. This study was conducted as a part of a mixed-method study [29].

### Study setting

The study was undertaken between September 2017 and February 2018 in the Khagrachhari district, CHT, Bangladesh. A detailed description of the larger study has previously been

reported [29]. The Khagrachhari district is hilly with a significant proportion of forested land [41]. This district was purposively selected because a range of public, private and non-profit health services were available to Indigenous women residing in the area, and because data on the Indigenous population were available [19, 42].

Two out of nine Upazilas (sub-districts), Matiranga and Khagrachhari Sadar, were selected based on availability of primary health care service centres to explore women's experiences of accessing MHC services. Three major Indigenous groups; Chakma, Marma and Tripura, inhabit the Khagrachhari district, comprising 50% (315,167 out of 613,917) of the district population and have distinctive cultural features including language [19, 42].

## Recruitment

The cross-sectional survey recruited 438 Indigenous women within 36 months of birth, who were asked permission to be contacted for further interviews based on their experiences of service use. Purposive sampling was used to recruit a diverse a group of women for interview representative of the population distribution in terms of reproductive age and ethnicity, including.:

i.  Users of any MHC service (ANC, facility delivery and/or postnatal care [PNC]), and

ii.  Non-users of MHC services.

## Data collection

Face-to-face, semi-structured interviews were conducted. Interviews were conducted by one female Bangladeshi researcher with an academic degree in Anthropology. The interviewer is non-Indigenous, but grew up in the region of the study site and has significant work and research experience with Indigenous women of the CHT, and with Indigenous populations globally. Interviews were conducted in the national language, Bangla.

Participants decided the time and location of the interviews, which were audio-recorded with consent. CHT Indigenous women work within and outside the home [19]. Interviews took place at the women's homes or in the fields or sites where the women were working, during work breaks as it was harvest season. The average interview duration was approximately 30 minutes.

Interviews sought to gather rich information regarding MHC service access until data saturation was achieved. Questions for the User group included:

i.  *Please describe your experiences in accessing antenatal/ delivery/ post-natal care services during your last pregnancy.*

ii.  *Why did you decide to access those services?*

iii.  *Which of the facilities did you find better and why?*

iv.  *What kind of barriers or problems did you face while accessing this service?*

Questions for the Non-user group included:

i.  *Please describe your experiences in accessing health care services in your community.*

ii.  *Why did you not access health facilities for services during your pregnancy?*

Socio-demographic characteristics and obstetric history were collected from each participant. Interview questions and consent forms were translated from English to Bangla by the

first author and reviewed by an independent researcher who spoke both languages and were approved by the relevant human research ethics committees.

## Data analysis

Interviews were transcribed directly into English and crosschecked by the research team. Data were coded through repeated review of transcripts linked to the research questions. Thematic coding captured emerging issues, logical connections and clarifications or comments that would help in explaining similar statements made by other participants. Analysis was driven by the data. Data categorisation included the identification of salient themes and sub-themes, recurring ideas and local meanings. Research team debriefings ensured the themes were reliable, following Braun and Clarke's [43] six step thematic analysis process. Qualitative data were analysed and organised systematically using a qualitative description approach aided by NVivo12 software [44, 45].

## Ethics

Ethical approvals were obtained from the Human Research Ethics Committee of the University of Newcastle, Australia (H-2017-0204) and the Department of Anthropology at Jagannath University in Bangladesh. Participation was voluntary, and confidentiality was maintained by using a pseudonym for interview participants. Results identify participants by a number only. Prior to accessing each *para* (village) for data collection, the interviewer contacted local community leaders of the Indigenous communities and sought their permission.

## Results

### Demographic characteristics of participants

Twenty-one Indigenous women were interviewed to reach data saturation; 14 MHC service users and seven non-users. The following socio-demographic data were collected:

**User group.** Among the 14 User group participants aged 18 to 35 years, five were Chakma, four were Marma and five were Tripura women. The majority attended school between primary and junior school levels (n = 9) with the remaining completing the secondary school certificate (n = 5). All women in the User group were homemakers; however, two were also day labourers, one a pre-primary schoolteacher, one a college student and 11 women were small shop owners. Most (n = 11) experienced their first pregnancy before their twentieth birthday.

Nine of the 14 User group participants accessed at least one public primary health care facility during their last pregnancy and childbirth, including Non-Government Organisation (NGO) run satellite clinics and government hospitals, located close to their homes. The remaining five accessed private facilities for either ANC (two women) or facility delivery services (three women).

For 11 participants, the nearest facilities were located between two and 10 kilometres from their homes. Facilities within two to five kilometres were accessible by walking or rickshaw (man-pulled three-wheel drive). Accessing facilities 10 kilometres away required a walk of up to one hour, then a wait for a bus or motorcycle. Often relatives would carry a pregnant woman with complications, a long-distance through hilly terrain, then hire a rickshaw to get her to the nearest hospital.

Only one participant accessed all three MHC services (ANC, facility delivery and PNC) during her last pregnancy. Six participants accessed ANC services and seven accessed facility delivery services during their last pregnancy. Although women who gave birth at a facility

received PNC services immediately after childbirth, participants were not aware of PNC services prior to this. Nine participants were pregnant once, and five participants were pregnant twice before when the interviews were conducted. A summary of users' socio-demographic characteristics and reported status of MHC service use during last pregnancy and birth is provided in Table 1.

**Non-user group.** Seven Indigenous women (three Chakma, two Marma and two Tripura women) who had not accessed MHC services participated in an interview. Five of these women had attended school between primary and junior school and the remaining two completed their secondary school certificate. Alongside their homemaker status, three participants worked as farmers (daily labourers) and one as a NGO community worker. Like the User group, most of the Non-user participants (n = 5) experienced their first pregnancy before their twentieth birthday. Participants of this group had not accessed any health facility in their previous pregnancies and deliveries, although most had a primary health care centre within two to five kilometres of their para (village). The majority of the participants had experienced more than one pregnancy. A summary of the non-user demographic characteristics and reported knowledge about MHC services is provided in Table 2.

**MHC services are for dealing with complications: "Why would I go, if I do not have any complications?".** Among the 14 User group participants, nine women had accessed ANC services; 11 women had accessed facility delivery services; and only one women had accessed PNC. Having complication at any point of pregnancy and childbirth was one of the key reasons for accessing MHC services. Of those women who accessed a facility delivery service, all but one did so due to complications. Complications included (as described by the participants) prolonged and obstructed labour, high blood pressure and first child had been delivered via

**Table 1. Socio-demographic and maternal health care (MHC) service use profile of the user group during last pregnancy and birth.**

| Socio-demographic information | | | | | | Information on MHC service knowledge and access | | | | | | |
|---|---|---|---|---|---|---|---|---|---|---|---|---|
| | | | | | | ANC | | | | Facility delivery | | PNC |
| Ethnicity | # | Age (*) | School attended & occupation | # of parity | Nearest health facilities & distance | ANC access | Reason for accessing or not accessing ANC | Total ANC visits | Trimester of 1st visit | Place & type of last delivery | Knowledge of facility delivery; source of information | Reasons for accessing or not accessing delivery service | Access status & reasons |
| Chakma | 1 | 18 (17) | Secondary; Homemaker | 1 | Satellite clinic and UHC; 3–10 km | Yes; Satellite clinic | • 1st pregnancy; • Check baby's condition | 3 | 1st | Home; vaginal birth | Yes; HW of NGO-run satellite clinic | • No complications • No one to take her to hospital | No Unaware |
| | 2 | 35 (24) | Secondary; Pre-primary School teacher | 2 | District hospital and private clinic; >20 km | Yes; Private clinic | • Relative was a doctor • Woman worked as HW | 3 | 2nd | Facility delivery; vaginal birth | Yes; health care provider of private clinic | • Past due date • Health care provider was relative | No Unaware |
| | 3 | 30 (17) | Primary; Homemaker | 3 | Satellite clinic and UHC; 5–10 km | Yes; HW visited home | • HW was a relative and lived nearby | 3 | 2nd | Facility delivery; vaginal birth | Yes; previous childbirth experiences at facility | • Previous bad experiences at childbirth • Health care provider was relative | No Unaware |
| | 4 | 30 (29) | Primary; Homemaker | 1 | CC and UHC; 5–10 km | No | • Aware but did not go • Perceived ANC would be expensive | 0 | — | Facility delivery; vaginal birth with episiotomy | Yes; neighbours had complications at childbirth | • Obstructive labour; • TBA unable to manage situation | No Unaware |
| | 5 | 20 (18) | 8th grade; Homemaker | 1 | UHC and District hospital 10–25 km | No | • Aware but did not go | 0 | — | Facility delivery; vaginal birth with episiotomy | Yes; health care provider | • To have safe childbirth | No; No problem faced |

*(Continued)*

**Table 1.** (Continued)

| Ethnicity | # | Age (*) | School attended & occupation | # of parity | Nearest health facilities & distance | ANC access | Reason for accessing or not accessing ANC | Total ANC visits | Trimester of 1st visit | Place & type of last delivery | Knowledge of facility delivery; source of information | Reasons for accessing or not accessing delivery service | Access status & reasons |
|---|---|---|---|---|---|---|---|---|---|---|---|---|---|
| | | | Socio-demographic information | | | ANC | | | | Facility delivery | | | PNC |
| Marma | 6 | 20 (19) | 9th grade; Homemaker & small business | 1 | District hospital and Satellite clinic; 8 km | Yes; Govt. hospital | • Had relatives at the hospital | 3 | 1st | Private facility; vaginal birth with episiotomy | Yes; health care provider | • High blood pressure; • Gynaecologist advice • Health care provider was known | No; No problem faced |
| | 7 | 20 (18) | Secondary; Homemaker | 1 | District hospital and Satellite clinic; 8 km | Yes; Satellite clinic and private clinic | • Availability of care providers; • Wanted to know the sex of the child | 4 | 1st | Facility delivery; C-section | Yes; previous childbirth experiences at facility | • First baby was C-section; • Went straight to the facility • Health care provider was known | Yes; Complications due Copper-T insertion |
| | 8 | 18 (16) | Primary; Homemaker & daily labourer | 1 | UHC; 5 km | No | • Unaware of ANC | 0 | – | Facility; vaginal birth with episiotomy | No prior knowledge | • Obstructed labour; • TBA unable to manage situation | No; Unaware |
| | 9 | 18 (17) | Primary; Homemaker | 1 | UHC and CC; 5 km | No | • Unaware of ANC | 0 | – | Private facility; C-section | Yes; neighbours had complications at childbirth | • Prolonged labour; • TBA unable to manage situation | No; Unaware |
| Tripura | 10 | 24 (16) | Primary; Homemaker | 2 | UHC and CC; 5 km | Yes; UHC | • Aware of ANC • HW visited home | 2 | 3rd | Private facility; vaginal birth | Yes; HW | • Prolonged labour; • TBA unable to manage situation | No; Unaware |
| | 11 | 19 (16) | Primary; Homemaker | 2 | UHC and CC; 5 km | Yes; Private clinic | • Went for ANC check due to accident and sickness | 3 | 2nd | Home; vaginal birth | Yes; neighbours had complications at childbirth | • Called a HW at home; • Feared expensive C-section at hospital | No; Unaware |
| | 12 | 29 (24) | Secondary; Homemaker | 2 | UHC and CC; 5 km | Yes; Private clinic; CSBA visited home | • Wanted to know the sex of the child; • Relative worked as CSBA and lived nearby | 4 | 2nd | Home; vaginal birth | Yes; neighbours had complications at childbirth | • Called a TBA • No complications; • Feared expensive C-section | No; Unaware |
| | 13 | 18 (16) | Primary; Homemaker & daily labourer | 1 | UHC and CC; 5 km | No | • Did not have complications | 0 | – | Facility delivery; vaginal birth with episiotomy | No prior knowledge | • Called a TBA • Prolonged labour; • TBA unable to manage situation | No; Unaware |
| | 14 | 20 (18) | Secondary; Homemaker & student | 1 | Satellite clinic and UHC; 5–12 km | Yes; Satellite clinic | • Relative worked as a HW and lived nearby | 4 | 1st | Facility delivery; vaginal birth with episiotomy | Yes; CSBA | • Prolonged labour; • CSBA unable to manage situation, referred to facility | No; Unaware |

* Age at first pregnancy; ANC = Antenatal care; CC = Community Clinic; CSBA = Community Skilled Birth Attendant; HW = Health Worker, TBA = Traditional Birth Attendant; PNC = Post-natal care; SSC = Secondary School Certificate; UHC: Upazila Health Complex

caesarean (C-section) birth. All of these women had tried delivering at home prior to accessing the delivery facility. Of those women who accessed ANC (n = 9), one had accessed ANC services

due to complications resulting from an accident during pregnancy. Finally, the only woman who accessed PNC services did so due to complications resulting from the insertion of a Copper-T. Among the Non-user group participants, women reported that they had not accessed MHC services because they had not experienced complications during pregnancy or childbirth.

This perception was common. One participant had accessed ANC checks during her last pregnancy and knew about possible pregnancy complications, yet chose to deliver at home with assistance from a traditional birth attendant (TBA) at childbirth.

> A pregnant woman can face many complications during delivery. . . if she can't deliver the baby (normally). . .or sometimes she could have heavy bleeding (haemorrhage). But, none of these happened to me! So, I did not go (to a facility)! I just had my labour pain, the *Ochai-Sung-Maa* [Tripura term for TBA] came and I delivered the baby! Why would I go, if I do not have any complication?" [User-TW-12].

Accessing facility delivery due to complications was perceived as bad luck for a woman.

> I think this is all about fate! If you are destined to have C-sectioning or '*side-kata*' (episiotomy), you have to have it. We believe that it is God's will" [Non-user-TW-07].

## Barriers that prevented women from accessing MHC services

For many of the participants there were multiple barriers that prevented them from even attempting to access MHC services.

**Lack of knowledge about the services and facilities.** Participants lacked knowledge about MHC services, particularly about ANC and PNC services. They had not attempted to access ANC and PNC services because they did not know about the services or their importance. One User and two Non-user participants were completely unaware of ANC services. Although 18 participants were familiar with ANC services, they were unaware of free or low-cost ANC services at community-based public facilities such as community clinics, Upazila health complexes or district hospital. Two User group participants, who had not accessed ANC services during pregnancy, resided within five kilometres of primary health care facilities and "no one ever told" them about the availability of such community facilities. Instead they had travelled more than 20 kilometres to access services at the nearest town centre. The majority of women believed that only private health facilities provided ANC services and that these services would be expensive.

> One of my friends had done her health check (at the private clinic) during her pregnancy. She told me it cost her a lot of money. [User-CW-04].

Another participant had accessed ANC services for her first pregnancy at a satellite clinic run by the United Nations Development Programme-Chittagong Hills Tract Development Facility (UNDP-CHTDF [a multi sectoral development program in partnership between United Nations Development Programs and the Government of Bangladesh]), which was no longer active at the time of data collection. She did not know that the same type of service was available at the local primary health care facility, free of cost, within three kilometres of her home. Instead, she assumed that she had to see a doctor at the hospital, which would be expensive.

> During my first pregnancy, I went to UNDP (-CHTDF's satellite) clinic near my parent-in-law's house. However, I did not go for any health checks for my last pregnancy as I wasn't aware of any such services nearby. [Non-user-TW-06].

**Table 2. Socio-demographic and maternal health care (MHC) services utilisation profile of the Non-user group during last pregnancy and birth.**

| Ethnicity | # | Age (*) | School attended & occupation | # of parity | Nearest facilities & distance | Ever accessed MHC services | ANC | Reasons for not accessing | Facility delivery | Reasons for not accessing | PNC |
|---|---|---|---|---|---|---|---|---|---|---|---|
| | | | | | **Socio-demographic status** | | | **Information on MHC service knowledge and access** | | | |
| Chakma | 1 | 22 (20) | Primary school; Homemaker and daily labourer | 2 | District hospital; >20 km | No | Yes | Husband did not take | Yes; neighbours had complications at childbirth | • relatives did not take; • no complications | No |
| | 2 | 22 (16) | Primary school; Homemaker and daily labourer | 2 | Satellite clinic and UHC; 5–10 km | No | Yes | Cost of private services | Yes; neighbours had complications at childbirth | • no complications | No |
| | 3 | 30 (19) | Secondary school; Homemaker, Community worker | 1 | UHC; 2 km | No | Yes | Family issues | Yes; Health worker | • no complications; • preferred home birth | No |
| Marma | 4 | 19 (18) | Primary school; Homemaker | 1 | UHC; 2 km | No | No | No complications | Yes; neighbours had complications at childbirth | • preferred home birth | No |
| | 5 | 22 (20) | Secondary school; Homemaker | 2 | CC and UHC; 3–8 km | No | Yes | Husband did not take | Yes; Health worker | • no complications • fear of episiotomy & C-section | No |
| Tripura | 6 | 24 (18) | 8th grade; Homemaker | 3 | CC and UHC; 3–8 km | No | Yes | CSBA checked once during 3rd trimester; Cost of seeing a doctor | Yes; neighbours had complications at childbirth | • preferred home birth; • no complications; • fear of episiotomy and C-section | No |
| | 7 | 20 (18) | Primary school; Homemaker and daily labourer | 1 | UHC 5 km | No | Yes | CSBA checked once during 3rd trimester; No complications | Yes; neighbours had complications at childbirth | • fear of episiotomy & C-section | No |

* Age at first pregnancy; ANC = Antenatal care; CC = Community Clinic; CSBA = Community Skilled Birth Attendant; PNC = Post-natal care; UHC: Upazila Health Complex.

One participant reported that she had wanted to see a doctor, but her husband did not take her as they were experiencing financial difficulties and were unaware of low-cost ANC-checks available at local primary health care facilities.

Of the 11 participants who had accessed facility delivery services, eight had accessed public facilities (district hospitals, Upazila health complexes and NGO-run satellite clinics) and three had accessed private facilities to give birth to their most recent baby. Two had a C-section and nine had a vaginal birth (six with an episiotomy). Three User group participants and all seven Non-user participants delivered their last child at home. Three User group participants had used ANC services but were not aware of delivery services. Both User and Non-user group participants reported that they came to know about facility delivery services from their neighbours who had experienced complications during childbirth (n = 9) and from health care providers (n = 8).

Of all MHC services, participants knew the least about PNC services. Although 11 out of 14 participants from the User group had delivered their last child at a facility, only one participant from this group reported accessing PNC services within one week of birth.

**Lack of knowledge on the importance of attending ANC and PNC services.** Among the 14 User group participants, 12 participants were aware of ANC services, but only nine had accessed ANC services during their last pregnancy. Six out of seven Non-user participants were familiar with the concept of ANC checks and knew it could provide information about pregnancy and the health of the baby. Relatives and neighbours, who were working as health care providers in a health organisation or had previous experiences of seeking health care during their own pregnancies, were the primary sources of information about ANC services. As a result of these informal information channels, most participants perceived that pregnant women should attend ANC services only if they experienced pregnancy complications.

I did not go (for ANC check) because I did not feel any complications during my pregnancy. If I thought I had any complications, I would have gone to see a doctor. I felt that as I was not having problems, why should I go then? [Non-user-TW-6].

Only one out of the 11 women who had delivered at a facility had been informed by staff about PNC services. They had only told to visit health care providers for follow-up if there were complications. Therefore, women had little understanding of the value of ANC and PNC services.

**Distance and lack of available transport.** Like many other Indigenous communities around the world, the Indigenous people of Bangladesh reside in remote areas where transportation is a substantial problem and many must travel considerable distances to access health services, often through hilly terrain. There could be a long wait to hire transport services and this was one reason that, for most women, a home birth was preferred.

Transport is a big issue for us. We have to wait quite a long time to get a transport. After reaching the hospital, nurses or doctors come late to see us. This delay causes us to suffer a lot. Because usually, we do not go to the hospital if the problem is not severe. When we arrive at the hospital, we are about to deliver the baby and the (labour) pain is at its peak! So severe!. . . but, nurses will be late to attend us. When they come, they just push saline and do not say anything. They will tell us to wait [Non-user-MW-05].

One participant had wanted to go to the NGO-run satellite clinic to ensure a safe birth; however, her water broke, and labour pains started before she could do so. The family was not prepared to deal with such a crisis. Within an hour, she delivered her baby.

I always feared to deliver at home; what if anything terrible happens! . . . On that day, I did not feel that much pain. My water broke first, and it did not take that long. . . I do not know (how long did it take) really! My husband went to arrange a transport but found nothing; meanwhile, the baby came out [User-CW-01].

**Fears related to medical intervention.** Seven participants expressed concern that if they attended a hospital facility for childbirth, they would be required to undergo a C-section, perceiving that staff would refuse to wait for normal vaginal birth. They were concerned that a C-section would be expensive for them and could be harmful for mother and baby.

In the hospital they do not allow doing normal (vaginal) delivery. They just do C-sectioning before the time and it cost me 12,000 taka (USD $142) for one C-section. But, I heard from people that it (C-section) causes problem. . .. I should have more than one. If I have my next baby through caesarean, it will be harmful for us (mother and baby) [User-TW-11].

Fears about facility delivery derived from relatives or neighbours with personal experiences in delivering their babies, or who had accompanied someone to a facility. Often the information was inaccurate. One participant, who accessed ANC services but not facility delivery services, reported that her neighbour had to go through a C-sectioning and the hospital staff then kept the baby in a specialised machine (incubator). After hearing this story, she perceived that facility delivery was painful and became scared of such services.

> It (facility delivery) was a harrowing experience, I heard. They keep the (newborn) baby inside a machine (incubator). The baby became so weak because of the C-section before the (expected) time. . .I don't know why they didn't wait for normal delivery or kept the baby in a machine [User-TW-11].

One Non-user Chakma participant, who lost her baby to pneumonia within one month of birth, feared that if she had accessed facility delivery services, she would have had to undergo a C-section. She also assumed that a pregnant woman would not get water to clean herself after giving birth in the facility, but that she would receive excellent care from her relatives if she stayed at home for childbirth.

> If you deliver at home, there will be no cut! Everything recovers so quickly. . . Because all my family members are here. . . they clean up everything with herbal things. . . wild plants. . . we also clean ourselves with hot water. It feels good as it reduces the pain quickly. . . (I think) you cannot do that in the hospital. . . they always (doctor) cut the belly. That is not good . . . they will give you medicine and take out everything (placenta) with devices or drugs. But, I do not know much about it clearly as I was not there before [Non-user-CW-03].

Participants also expressed fears about a range of medical interventions such as injections or saline that might be used at the facility to induce labour, which were considered unnecessary and risky. C-sectioning was perceived as dangerous because doctors used a sharp instrument and it was feared that this could harm the unborn baby.

> In the hospital, they will push saline and injections. If they fail to control the situation, they will do the C-sectioning and that is even worse and risky! It can harm the baby as they use sharp things doing the C-section! [User-TW-12].

Another Non-user participant mentioned she heard that facility staff would randomly check the patient's vagina with their hands, with the excuse of monitoring the child's position.

> I have not been to a hospital for delivery. What I have heard from my neighbours that they (staff) put their hands (in vagina) randomly to check the child's position. They do not listen to you (the patient) even if you tell them [Non-User-CW- 02].

Concerns about being forced to walk for a long time and undergoing *side-kata* (episiotomy) were also frequently mentioned by the participants.

> I fear (*dor laagey*) going to hospital. I have learnt that the hospital staff will make you walk a lot if you deliver at the hospital and they will do '*side-kata*' if the baby's size is large. If the delivery takes a bit longer, they will go for C-sectioning! I fear that! [Non-user-TW-07].

To avoid any medical interventions (such as random vaginal checks), pregnant women reported preferring to stay at home to give birth, where their privacy could be maintained.

> To me, during delivery, every woman feels pain; severe pain. It is normal. You have to tolerate and accept the pain, as it does not stay long. If you keep patience, you will deliver your baby soon. . . only a few people will be around. . . you can have better privacy at-home delivery. I prefer this [Non-user-TW-06].

## Deterrents to future MHC service access

The barriers to accessing MHC services, particularly public MHC services, were also founded on personal previous experiences of health services, reflecting women's perceptions of failures and weaknesses of the health system and services.

**Lack of trust in the quality of care at public facilities.**   One Marma participant, a first-time mother, mentioned that safe motherhood was more important to her than cost. She knew that ANC services at government facilities could be accessed free of charge. However, her experience was that the government hospital was overcrowded, with a long queue of patients due to poor management. Instead, she chose to see the same doctor (who also worked at a government hospital) at a private facility.

> "I know that the doctor (name) sees patients both at the hospital and private clinic. Everyone goes to Sadar hospital that is always overly crowded. The management is not good enough. There is no proper queue to see the doctor. Sometimes you won't get a place to sit. My tummy became so big and my feet and palms were swollen that I could not wait long to see the doctor at the hospital. So, I had no choice except going to the (private) clinic and did not think about money then" [User-MW-06].

**'Unofficial' costs–Even in so-called 'free' facilities.**   Most participants who accessed delivery services described a range of out-of-pocket expenses for travel, medicine, accommodation and food. In addition, many noted they had had to pay hospital staff extra money for their services. Most participants were unaware that normal vaginal birth is free in all public facilities [46]. All participants who accessed a service for vaginal birth were asked to pay "tips–*Bakhshish*" (gift–incentive) to the facility staff, including cleaning staff. This payment was considered as a way "to make them (the staff) happy" [User-MW-03].

> I had to pay 2000 taka (USD 23.7) to them (hospital staff). They demanded this from us. The *aaya* (hospital cleaner) came to me and asked for money as she had cleaned my clothes. As far as we know everybody who goes to the facility, has to pay them this amount of money or more. They directly ask the patient for it. [User_CW-05].

One Tripura participant from the User group reported having a normal vaginal birth at the Upazila health complex where staff asked for money from her relatives. Staff asked for money from all patients who delivered at the hospital regardless of their socio-cultural status and considered "*bakhshish*" a normal practice (*Ditei hoy*–you should give them) "but it is not the hospital cost" [USER-TW-14]. There was no fixed amount. Requests ranged between 2000 and 3500 Bangladeshi Taka (USD $18 to 42) for a standard vaginal birth. Often they bargained with patients' relatives according to their ability to pay. This *bakhshish* would buy better services from the staff.

The hospital staff behave well to people who have money. They give better services to rich people. Those who do not have money, they are most likely to be neglected by the staff. They always shout at those people [User-MW-07].

As with the User group, Non-user participants who were aware of facility delivery services, were unaware of the free or low-cost services for standard vaginal birth at the nearest public and NGO-based satellite clinics. However, they too had heard about the unofficial costs for services at the public facility, mainly from their neighbours and relatives. One Non-user participant had accessed facility delivery services immediately after giving birth to her first child a few years ago. She had delivered her baby at home, but experienced pain and bleeding due to tearing and went to a hospital where nurses demanded payment for suturing. From this experience, she decided not to access facility services for her next childbirth.

They did not start stitching my torn part; they demanded 1000 taka (USD $12) for that. But we bargained, as we did not have money. Then, my mother gave them 500 taka (USD $6) [Non-user-CW-02].

**Maltreatment from hospital staff.**   Maltreatment from staff was identified as a deterrent to accessing facility delivery services. The majority of User group participants complained that most government facility staff, including medical, administration and cleaning staff, behaved badly with the patients.

I did not like the way they shout at people all the time... You know that at the hospital there will be patients with a different type of problems. A medium noise may cause discomfort to the patient, feels so loud for the patient . . . they do not speak in normal tone or care about the patient's discomfort [User-MW-06].

Another User group participant reported that the doctor had inserted an intra-uterine device for contraception without prior discussion. The participant came to know about the insertion while checking out of the facility as she was feeling discomfort and pain. She expressed frustration with the doctor and staff, as they had not explained the device's side effects.

They put Copper-T (intra-uterine device) in during the C-section. The doctor put it in without asking me. . . . She (the doctor) told my husband when we were about to get released from the hospital and I had no idea about what a Copper-T was. I felt bad. I would not have felt that bad if they did it with my consent [User-MW-07].

Non-user participants had witnessed maltreatment while attending health facilities with other patients or had heard about such maltreatment from their relatives or neighbours. One Non-user group had seen staff, including Indigenous staff, scolding and shouting at pregnant women and relatives on previous visit to the health facility.

*Didi* (sister), they scolded and shouted at us a lot! That's what I find very bad. Not only the nurse, but also the cleaners mistreated us [Non-user-CW-02].

This Non-user participant had also attended the nearby government facility and received aggressive treatment from the staff for the tear she had sustained during home birth.

The nurses checked and told me that the tearing was bad. . . They took off my clothes without asking me and scolded us badly. I felt bad and did not access their services again [Non-user-CW-02].

The quality of treatment was believed to depend on how familiar the patient or her family members were with the hospital staff and the level of wealth of the patient's family. A few User group participants mentioned that they had relatives who worked as health care providers at the facility and they were treated well by the staff.

We knew the staff there. So, they [staff] treated us well. But, those who do not know anyone in the hospital, the hospital staff mistreat them [User-MW-07].

**Lack of appropriate and accessible information.**   One Non-user participant reported that hospital staff neither provided clear information about the delivery procedure nor explained childbirth-related health problems. When they did they tended to use technical terms and it was difficult for patients and their relatives to understand.

I think we do not understand the importance of the service; we do not receive health education correctly. . . I believe this is why we don't access these services. . . this (delivery procedure) is not clear to others. . .health care providers should make it clear to us and I think it would be better if they use our language. It is crucial to passing these health messages using our language or terms that we can understand [Non-user-MW-05].

Participants reported that they were not aware of PNC services. The only participant, who access PNC service, had complications due to insertion of a Copper-T without consent. The other 10 participants stayed at the facility for one or two days after delivering their baby, depending on the extent of complications. They could not tell whether they had received PNC services during their stay at the facility, as they were not aware of this category of MHC service. While checking out from the facility they were told to visit health care providers in case of any complications, but they did not know what those complications might be.

One participant, who had experienced an episiotomy, said that the hospital staff told her how to take care of her stitches. The staff advised her to attend a follow-up visit but did not explain why it was essential to see a health care provider within six weeks of childbirth. Therefore, she did not go, as she felt "fine".

They prescribed me *Sanora* (sanitary napkin brand), ointment for my cut. . .they gave the vaccine to my baby. Then they told me to avoid heavy work, to drink a lot of water and to eat healthy foods. They told me to go and see them after one week if I faced any complications. But, I did not go as we, my child and I, were fine [User-MW-06].

Another Marma participant reported attending a facility and needing *"side-kata"* (episiotomy) to deliver the baby. Facility staff told her how to care for the stitches but did not tell her about PNC services.

They only told me to walk carefully and to clean my delivery passage. . .they told me to keep the cut–stitches (for episiotomy) clean after going home. . . They did not tell me anything about how to breastfeed the baby, not even about vaccination. I did not go to the hospital as I did not face any problems [User_MW_08].

Four participants reported that facility staff did not tell them about PNC services, but they went to a facility to vaccinate their newborn babies. They learnt about the importance of child vaccination from community health workers and relatives.

## Maternal health care values

All participants said that they valued door-step MHC services, felt more comfortable at home than at facilities and preferred the help of TBAs to facility staff. These preferences offer important insights into the factors that might better enable women's access to MHC services.

**The importance of home.**   All participants discussed their preference for staying at home during pregnancy and childbirth. According to one Non-user participant, hospital was a place where people go only when they have "*somossya*" (problems or health-related issues).

> We all prefer home birth. Previously there was no hospital in this area. Hospital is a new version of health care service. We did not have a hospital at an early age. Our elders delivered at home. So we prefer delivering at home if the delivery happens normally. In case of "*somossya*", we go to a hospital [Non-user-CW-01].

Doorstep services were the preferred model of service delivery. Under the UNDP-CHTDF development program, health workers visited women's homes to provide doorstep services, including ANC checks. According to participants, such services stopped suddenly in 2016.

> They [UNDP] used to provide free doorstep services to us a few years back. They suddenly stopped their services. . . My mother used to work for UNDP as a health worker. So, I know about it. (ANC check). [User-TW-14].

After the closure of the UNDP-CHTDF interventions, people in the intervention areas seemed unaware of available MHC services. All seven Non-user participants and three User participants were unaware of ANC services available from facilities located within three to 10 kilometres. Another Non-user Chakma participant reported that the Union health complex was the closest facility to her home, but it was not sufficiently staffed and did not provide door-step services.

> We have a Union health centre nearby but it is not well set up. Health staff come to the centre from Matiranga only on Saturday and Tuesday. . . But no doctor comes there. I have not been there yet [Non-user-CW-02].

Another Non-user Marma woman had not attended ANC checks due to the lack of door-step services. She suggested that the government should increase doorstep services as they enabled health promotion and encouraged Indigenous women to access services.

> The government should increase home-visits. If government health offices like Family Planning Office send health workers to each *para* three to four times in a week and promote health education in a way that we can understand and make it clear to us, then I am sure, women will feel encouraged to go to the facilities [Non-user -MW-05].

**Flexible payment options: "We can pay them (TBAs) later".**   Seeking help from a local TBA was a common practice among Indigenous women, even though community skilled birth attendants (CSBA) were available closer to their villages. Indigenous women usually sought help from untrained TBAs first, as they were considered more accessible and

affordable. CSBAs used medical items such as gloves, blades, injections or saline, whereas TBAs would use oil and warm water and sometimes herbal medicines.

> The *sharama* (Marma term for TBA) first told me to lay down and massaged oil on my belly. She also pushed (belly) for a while, gave me warm water to drink and some herbal medicine prepared from wild plants [Non-user-MW-04].

Participants from the Non-user group identified that costs associated with TBAs made them an accessible option as there was no fixed payment, and they did not have to pay upfront.

> You can pay her according to your affordability. You can pay her later as well. However, it was less than what it costs in the hospital [Non-User-CW-02].

In comparison, patients had to pay out-of-pocket expenses immediately after accessing facility delivery services. Family members of the Indigenous women did not have money saved for unexpected complications of pregnancy and childbirth and so often had to access a loan or mortgage their properties to pay for facility services.

> I do not know (the total cost). My mother-in-law knows it. Nevertheless, I know that we did not have money. My mother mortgaged her (gold) earring and managed money to pay the hospital cost [User-MW-08].

**Personal connections.**   Six participants noted that having relatives working as health care providers had encouraged them to access ANC services.

> One of my uncles is a doctor. He suggested me to go to this doctor female doctor [name] [for ANC check], and I also know her personally. [User-TW-12].

Personal connections also influenced Indigenous women's relatives including husband to make decision whether they would take the woman to any facility for childbirth. Four participants reported that they accessed facility delivery services directly because they had personal connections with the health care providers (two had relatives and two had personal friendly relations with health care provider). As a result, husband and other family members were well aware of the importance of delivering at facility. One participant mentioned that her husband took her directly to a hospital where their relative worked as a doctor.

> The doctor [name of the doctor] is my husband's relative. . . They are cousins. [User-CW-02].

## Discussion

This qualitative study reports perspectives of Indigenous women in the CHT, a group who have previously had little or no voice, in order to address inequalities in MHC access and health outcomes. This qualitative data offers an in-depth understanding of the barriers that prevent the women from accessing MHC services, experiences that shape their reluctance to access formal MHC services and characteristics that they value in MHC.

Like Indigenous women in Guatemala, Mexico and Vietnam, Indigenous women in the CHT experienced significant inequality and inequity regarding accessing MHC services

during pregnancy and at childbirth [36–38]. However, unlike these countries, no targeted intercultural health care service for Indigenous women was found in the CHT and often these women experienced discrimination (including unofficial payments and abusive treatment) from health staff, including Indigenous health staff, at the public facilities. As with previous studies, this study found that Indigenous women lacked knowledge about existing primary health facilities and about the importance of accessing MHC services [36–38]. These findings are supported by literature that state the importance of relatives and neighbours, may provide incomplete information regarding ANC and facility delivery [47–49]. One unique finding of the study was that having personal connection with health care providers influenced Indigenous women and family members' decisions regarding accessing ANC and facility delivery services.

Indigenous women in the CHT reported that pregnancy was a natural event. MHC services were perceived to be essential to access only if a woman was experiencing "*somossya*" (complications) during pregnancy and at childbirth, when situations became uncontrollable and were beyond the capacity of a TBA [36, 37, 48, 50, 51]. Otherwise, MHC services were considered a waste of time and money. Furthermore, having medical conditions played a key role for Indigenous women who accessed facility delivery services. Unlike previous literature, this study found neither travel time nor distance were major barriers to accessing MHC services [52]. This could be because Indigenous women had limited knowledge about existing facility services within their communities and therefore, had not considered the distance [29].

CHT Indigenous women, including service users, perceived that hospital staff have a preference for performing "*pet-kata*" (C-section) or "*side-kata*" (episiotomy). Indigenous women expressed their fears regarding such medical interventions and saw these as unnecessary, harmful and expensive. Such misconceptions indicate that Indigenous women did not receive appropriate health knowledge when they accessed MHC services. Not having comprehensive health knowledge about modern MHC interventions and led to lower rates of MHC usage among Indigenous women [53, 54]. This was the case for Indigenous women in the CHT, despite basic MHC services being free at public health facilities in Bangladesh [21]. Given that word-of-mouth is an influential channel of communication, it is likely that this type of information will need to be shared and disseminated in face-to-face modes.

Findings of this study show that having Indigenous health care providers in public facilities did not change the scenario for Indigenous women in the CHT [15, 55]. Women, regardless of ethnic identity, faced abusive and poorer quality treatment from staff if they failed to pay unofficial fees. Considering this unofficial payment as a "gift" (*Bakhshish*) or "buying quality care" to avoid maltreatment from the staff was a way of justifying such unethical and unfair demands. Furthermore, Indigenous women and their relatives perceived such payment was normal [55, 56]. Arranging money for transportation, accommodation and medicines was already a major out-of-pocket expense for rural Indigenous people [33], and this additional cost was an extra burden [56]. Such illegal and forceful payment (in the name of "gift") in the maternity care sector contributes to a sense of distrust and disrespect towards the health system [56].

Calling a TBA to support childbirth was the first preference due to accessibility, affordability, availability [36, 37] and perceived trustworthiness. The option of paying the TBA later was convenient and cost-effective for poorer Indigenous families. Free doorstep services provided by health workers positively influenced Indigenous women to access MHC services, particularly ANC and facility delivery services [24, 57]. Given the nature of Indigenous women's work within and outside their households, findings of this study reinforce the importance of free doorstep services for this population.

There seemed to have been little done to ensure Indigenous people were connected with local free or low-cost health services after the closure of the UNDP-CHTDF program in 2016 [24]. This reiterates findings from other low and middle-income countries studies that show a lack of sustainable approaches to health care funding and service provision for Indigenous women [32, 36]. Indigenous women responded well to doorstep MHC approaches by Indigenous health workers, provided that they were delivered in a culturally sensitive and appropriate manner. However, programs such as UNDP-CHTDF must be delivered with consideration to sustainability [33]. There was a missed opportunity in the closure of the UNDP-CHTDF program to connect women with mainstream public MHC services. CSBAs from Indigenous communities who became unemployed after completion of UNDP-CHTDF program could be employed as a resource at the local community level.

Engaging Indigenous communities from the beginning has been shown to improve understanding, acceptability, ownership and sustainability of health programs [37]. Having ethnically diverse health staff, including doctors, nurses and CSBAs, working in health facilities (public, private and NGO-based) in the CHT [24, 58] was assumed to provide culturally competent health care [48, 59]. However, like previous findings from LMICs, maltreatment from facility staff, including Indigenous health staff, was an important barrier to accessing facility delivery for Indigenous women [33]. Women reported that experiences with MHC services had felt invasive, intrusive and unwelcoming and that this led to feeling uncomfortable, fearful and distrustful of public services [36]. Having Indigenous staff working at the facilities was insufficient on its own in bridging cultural gaps and barriers and there were further missed opportunities to educate Indigenous women (particularly by Indigenous health staff) on MHC services. Subsequent studies should seek to understand the type of training and structures required to improve cultural appropriateness, build trust and disseminate reliable information for Indigenous women on MHC services [52].

These qualitative findings suggest that CHT Indigenous women's low levels of access to MHC services are due to lack of knowledge about the importance and value of these services and their concerns about the associated costs and methods of treatment. Ultimately, women perceived that they could receive more culturally appropriate, private, accessible and affordable care via private providers and TBAs in their home settings.

Measures to improve Indigenous women's access to MHC services in the CHT must recognise structural and systemic weaknesses. Accountability measures need to be introduced to address the widespread illegal practice of "*bakshish*" (gifting) within public health services. Such practices further exclude marginalised, Indigenous communities. Intercultural policies and interventions that support human rights and gender equity are needed to access hard-to-reach population groups such as the Indigenous women living in the CHT. Furthermore, health staff, including Indigenous, administration and cleaning staff, need to undergo cultural competency training and monitoring to ensure a culturally safe environment for Indigenous women, where communication is considerate, transparent and compassionate.

## Strengths and limitations

Recruiting participants from three major ethnic groups [19] has made this study unique. Checking of interview transcripts against study objectives, triangulation of codes with co-authors and feedback of themes were used to ensure rigour. To our knowledge, this is the first published qualitative study exploring Indigenous women's perspectives on barriers to accessing MHC services in the CHT. Capturing the voices of Indigenous women from geographically disadvantaged areas regarding their lived experiences of accessing MHC services is the key strength of this study. This is pioneering research in developing an understanding about

inequalities and inequities in Indigenous women's health. Purposeful sampling was used to select a wide range of service users and non-users with different experiences [60].

The main limitation is that a small sample of women was used for this study which limits transferability of findings outside the CHT. However, as the health care delivery system and inequities in access are similar for Indigenous communities of Bangladesh, the results might have broad applicability in similar environments. Limited data on enablers of health service use were determined despite having two questions in the interview schedule for service users to elicit enablers; however the focus from the women was on barriers to access.

Another potential limitation of this study was not using Indigenous native languages for the interview, particularly with participants with less education, which may have led to communication barriers [61]. However, this is one of the first studies to be conducted in Bengali. Interviewers' limited knowledge of Indigenous terms for pregnancy-related complications might have limited data collection. Privacy of interviews may have been compromised given the family structure and because Indigenous participants often felt more comfortable with relatives around, the researcher was aware of this dynamic but no participants raised this as a difficulty [24].

## Conclusion

This qualitative study provides an in-depth understanding of barriers to accessing MHC services for CHT Indigenous women. Findings highlight the multi-faceted changes required to improve MHC access for Indigenous women; from health education, to making changes to policy and interventions. There are currently missed opportunities that could be utilised to build women's knowledge about accessing MHC services. Despite having Indigenous health staff at community level and at facility centres, the environment was not culturally acceptable for Indigenous women. Culturally appropriate and user-friendly information about MHC needs to be shared from pre- to post-pregnancy, including their importance, availability and means of accessing free MHC services. Indigenous women also reflected on their preferred MHC services by emphasising their preferences for doorstep services and flexibility in payment. The barriers to access reflect the important cultural values, priorities, fears and concerns of women and their communities about health, health services and the health services system. To understand and address these barriers, health promotion and intervention programs must be designed in collaboration with Indigenous communities. Reform is required at individual, community and health systems levels to enable Indigenous women from the CHT to make informed choices and empower them to birth their babies safely according to their personal, cultural and social preferences.

## Acknowledgments

The authors are indebted to all the Indigenous women who took part in the study and shared their information for this research, and local community leaders for their invaluable support by approving the access to communities. Special thanks to the field research assistants for their incredible work during data collection.

## Author Contributions

**Conceptualization:** Shahinoor Akter.

**Data curation:** Shahinoor Akter.

**Formal analysis:** Shahinoor Akter.

**Methodology:** Shahinoor Akter.

**Software:** Shahinoor Akter.

**Supervision:** Kate Davies, Jane Louise Rich, Kerry Jill Inder.

**Writing – original draft:** Shahinoor Akter.

**Writing – review & editing:** Kate Davies, Jane Louise Rich, Kerry Jill Inder.

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
