## [Decision Letter · Decision Letter 0]

6 Apr 2020

PONE-D-20-03726

Indigenous women’s perspectives on accessing maternal health care services in the Chittagong Hill Tracts (CHT), Bangladesh: a qualitative descriptive study

PLOS ONE

Dear Shahinoor Aktor,

Thank you for submitting your manuscript to PLOS ONE. After careful consideration, we feel that it has merit but does not fully meet PLOS ONE’s publication criteria as it currently stands. Therefore, we invite you to submit a revised version of the manuscript that addresses the points raised during the review process.

We would appreciate receiving your revised manuscript by 6th May 2020. To enhance the reproducibility of your results, we recommend that if applicable you deposit your laboratory protocols in protocols.io, where a protocol can be assigned its own identifier (DOI) such that it can be cited independently in the future. For instructions see: http://journals.plos.org/plosone/s/submission-guidelines#loc-laboratory-protocols

We look forward to receiving your revised manuscript.

Kind regards,

Sharon Mary Brownie

Academic Editor

PLOS ONE

Journal Requirements:

2. When reporting the results of qualitative research, we suggest consulting the COREQ guidelines: http://intqhc.oxfordjournals.org/content/19/6/349. In this case, please consider including more information on the number of interviewers, their training and characteristics.

Additional Editor Comments (if provided):

Reviewers have returned a range of recommendations aimed at improving your manuscript. Please consider each recommendation and revise your manuscript accordingly. Please revise within the manuscript and also submit a table showing how you have addressed each of the recommendations.

Reviewers' comments:

Reviewer's Responses to Questions

**Comments to the Author**

1. Is the manuscript technically sound, and do the data support the conclusions?

Reviewer #1: Yes

Reviewer #2: Partly

2. Has the statistical analysis been performed appropriately and rigorously? 

Reviewer #1: N/A

Reviewer #2: N/A

3. Have the authors made all data underlying the findings in their manuscript fully available?

Reviewer #1: Yes

Reviewer #2: No

4. Is the manuscript presented in an intelligible fashion and written in standard English?

Reviewer #1: Yes

Reviewer #2: No

5. Review Comments to the Author

Reviewer #1: Comments on Manuscript: Indigenous women’s perspectives on accessing maternal health care services in the Chittagong Hill Tracts (CHT), Bangladesh: a qualitative descriptive study

Thank you for requesting me to review this manuscript. This paper is an important contribution to the use of MCH services and disrespect and abuse of women in MCH services by health workers. A lot of this work has been done in low (e.g. Malawi) and middle income countries. However, this paper is important because it focuses on marginalized communities.

Introduction

Perhaps it would be proper to indicate how many ANC visits WHO expects women to have.

Analysis

1. Was the analysis theory or data driven?

2. The analysis could benefit from further categorization or classification of the barriers according to various themes (e.g. health care barriers, individual barriers, community barriers etc.). The Bradley paper below could perhaps give some insights on how to arrange the themes.

A) System barriers - could look at those barriers related to the health care systems

B) Health provider barriers e.g. unethical practices that are done without informed consent of the participants who have a loop inserted without their knowledge, midwives and cleaners who request for payment from patients

C) Individual barriers - These would pertain to the barriers pertinent to the participants like lack of money, lack of knowledge etc.

Some categories were not necessarily barriers, but merits of the home care delivery, so maybe they could fall under perceptions of participants, like, the comparison they make between health care and home delivery.

I would add to line 655 - unethical and unfair demands

References

Bradley, S., McCourt, C., Rayment, J., & Parmar, D. (2016). Disrespectful intrapartum care during facility-based delivery in sub-Saharan Africa: a qualitative systematic review and thematic synthesis of women's perceptions and experiences. Social Science & Medicine, 169, 157-170.

Reviewer #2: Identification of barriers to access of maternity care is important. It is disappointing that the researchers did not look at enablers as well, as often the voice of indigenous women about what has worked/is working well identifies what can be strengthened. The barriers identified are generalised to other indigenous women not only to other areas in Bangladesh but worldwide. I would like to suggest that the authors revisit these assumptions, in particular as purposive sampling was used with a small group of women from CHT region. It is unclear if the participants all spoke the same language or whether Bangla was their first language. Having information or service provision in one's own language was not identified as a barrier in the abstract and received only a small menton in the article. Numerous health literature research has shown that this is one of the biggest barrier. Some sentence structures are long and lose clarity. There is inconsistency with some terminology, e.g. Bangladesh is a low to middle income country but sometimes referred to as low income country. It appears the author assumes being indigenous means diverse socio-cultural backgrounds. Clarity around this would enhance the understanding for the reader. It would be useful to include a short paragraph explaining the difference between minority groups and indigenous groups and describe the general indigenous people group of Bangladesh. It appears the term minority groups and indigenous groups are used interchangeable in this study. The corresponding article with this study published by BMJ: Akter S, Rich JL, Davies K, Inder KJ. Access to maternal health care services among Indigenous women in the Chittagong Hill Tracts, Bangladesh: A cross-sectional study. BMJ Open. 2019. doi: http://dx.doi.org/10.1136/bmjopen-2019-033224 is well written and it may be useful to enlist the support of the co-authors of that study to address the long sentences and repeat expressions in the current article.

Other studies who adress barrier identifications have also asked the participants how to overcome these. If the current authors plan further studies in this area, it is suggested to consider this as an inclusion.

6. PLOS authors have the option to publish the peer review history of their article (what does this mean?). If published, this will include your full peer review and any attached files.

Reviewer #1: No

Reviewer #2: No

---

## [Author Response · Author response to Decision Letter 0]

14 Jun 2020

A rebuttal letter has been uploaded separately (labeled as 'Response to Reviewers') in response to each point raised by the academic editor and reviewer(s).

---

## [Decision Letter · Decision Letter 1]

4 Jul 2020

PONE-D-20-03726R1

Barriers and enablers to accessing maternal health care services in the Chittagong Hill Tracts, Bangladesh: A qualitative descriptive study of Indigenous women’s experiences

PLOS ONE

Dear Dr.Shahinoor Akter,

Thank you for submitting your manuscript to PLOS ONE. After careful consideration, we feel that it has merit but does not fully meet PLOS ONE’s publication criteria as it currently stands. Therefore, we invite you to submit a revised version of the manuscript that addresses the points raised during the review process.

We look forward to receiving your revised manuscript.

Kind regards,

Sharon Mary Brownie

Academic Editor

PLOS ONE

Additional Editor Comments (if provided):

Reviewers have recommended some further areas for improvement within your manuscript. Please respond to each and every recommendation and return your revised manuscript along with a table of author responses.

Reviewers' comments:

Reviewer's Responses to Questions

**Comments to the Author**

1. If the authors have adequately addressed your comments raised in a previous round of review and you feel that this manuscript is now acceptable for publication, you may indicate that here to bypass the “Comments to the Author” section, enter your conflict of interest statement in the “Confidential to Editor” section, and submit your "Accept" recommendation.

Reviewer #1: (No Response)

Reviewer #2: (No Response)

2. Is the manuscript technically sound, and do the data support the conclusions?

Reviewer #1: Yes

Reviewer #2: Yes

3. Has the statistical analysis been performed appropriately and rigorously? 

Reviewer #1: N/A

Reviewer #2: N/A

4. Have the authors made all data underlying the findings in their manuscript fully available?

Reviewer #1: Yes

Reviewer #2: No

5. Is the manuscript presented in an intelligible fashion and written in standard English?

Reviewer #1: Yes

Reviewer #2: Yes

6. Review Comments to the Author

Reviewer #1: Most of the comments have been adrressed. The last sentence can be shortened to read: Although there were two questions meant to elicit enablers of health service use, the data were limited as women focused more on barriers to access.

On page 4 line 97 there is an APA reference which should be corrected to align with the journal referencing.

There is a minor editorial on page 27 line 561, delete a.

Otherwise with these minor changes, the paper can be accepted.

Reviewer #2: It is commendable that the author has addressed most of the review feedback provided. Here are further recommendations to strengthen the article:

- although it was recommended that the title changes to barrier and enablers..., there however are no enablers identified in the re-submitted article as suggested, hence I recommend now to remove the word 'enabler' from the title

- line 148 remove the word facilitators, as they are not identified in the article

- line 168 'upazilas' needs a capital U

- line 242 please add women after 'Most (n=11)'

The abstract conclusion and the article conclusion do not really reflect the findings. 'What is meant with 'systemic reform' ?- Revisit the following article and https://bmjopen.bmj.com/content/9/10/e033224 and read the conclusions and abstract. It is informative to see that the language concern is being mentioned in the abstract and this additions is welcomed and importunate, however, it features only marginally in the study, hence other significant findings need to be mentioned e.g. fear, distance, costs, lack of trust, maltreatment from staff, lack of knowledge all are important barriers

- Please also note on the reference list: reference 38 and reference 39 are missing their doi's. Either add them or delete doi

- Table 1 - #3 and #4 while km has now been added there needs to be gap between the number and km as it has been presented with all others, consistency is required. Thank you.

7. PLOS authors have the option to publish the peer review history of their article (what does this mean?). If published, this will include your full peer review and any attached files.

Reviewer #1: No

Reviewer #2: No

---

## [Author Response · Author response to Decision Letter 1]

12 Jul 2020

A rebuttal letter titled "Response to Reviewers" has been uploaded in response to specific reviewer and editor comments.

---

## [Editor Report · Decision Letter 2]

20 Jul 2020

Barriers to accessing maternal health care services in the Chittagong Hill Tracts, Bangladesh: A qualitative descriptive study of Indigenous women’s experiences

PONE-D-20-03726R2

Dear Dr. Shahinoor Akter,

We’re pleased to inform you that your manuscript has been judged scientifically suitable for publication and will be formally accepted for publication once it meets all outstanding technical requirements.

Kind regards,

Sharon Mary Brownie

Academic Editor

PLOS ONE

Additional Editor Comments

Reviewer recommendations have been satisfactorily addressed.

---

## [Editor Report · Acceptance letter]

30 Jul 2020

PONE-D-20-03726R2 

Barriers to accessing maternal health care services in the Chittagong Hill Tracts, Bangladesh: A qualitative descriptive study of Indigenous women’s experiences 

Dear Dr. Akter:

I'm pleased to inform you that your manuscript has been deemed suitable for publication in PLOS ONE. Congratulations! Your manuscript is now with our production department. 

Kind regards, 

on behalf of

Professor Sharon Mary Brownie 

Academic Editor

PLOS ONE